# Answerer in Questioner's Mind: Information Theoretic Approach to Goal-Oriented Visual Dialog

**Sang-Woo Lee**[1*]**, Yu-Jung Heo**[2]**, and Byoung-Tak Zhang**[2,3]

Clova AI Research, Naver Corp[1]
Seoul National University[2]
Surromind Robotics[3]

## Abstract

Goal-oriented dialog has been given attention due to its numerous applications in artificial intelligence. Goal-oriented dialogue tasks occur when a questioner asks an action-oriented question and an answerer responds with the intent of letting the questioner know a correct action to take. To ask the adequate question, deep learning and reinforcement learning have been recently applied. However, these approaches struggle to find a competent recurrent neural questioner, owing to the complexity of learning a series of sentences. Motivated by theory of mind, we propose "Answerer in Questioner's Mind" (AQM), a novel information theoretic algorithm for goal-oriented dialog. With AQM, a questioner asks and infers based on an approximated probabilistic model of the answerer. The questioner figures out the answerer's intention via selecting a plausible question by explicitly calculating the information gain of the candidate intentions and possible answers to each question. We test our framework on two goal-oriented visual dialog tasks: "MNIST Counting Dialog" and "GuessWhat?!". In our experiments, AQM outperforms comparative algorithms by a large margin.

## 1   Introduction

Goal-oriented dialog is a classical artificial intelligence problem that needs to be addressed for digital personal assistants, order-by-phone tools, and online customer service centers. Goal-oriented dialog occurs when a questioner asks an action-oriented question and an answerer responds with the intent of letting the questioner know a correct action to take. Significant research on goal-oriented dialog has tackled this problem using from the rule-based approach to the end-to-end neural approach [1–3].

Motivated by the achievement of neural chit-chat dialog research [4], recent studies on goal-oriented dialogs have utilized deep learning, using massive data to train their neural networks in self-play environments. In this setting, two machine agents are trained to make a dialog to achieve the goal of the task in a cooperative way [5–7]. Many researchers attempted to solve goal-oriented dialog tasks by using the deep supervised learning (deep SL) approach [8] based on seq2seq models [9] or the deep reinforcement learning (deep RL) approach utilizing rewards obtained from the result of the dialog [10, 11]. However, these methods struggle to find a competent RNN model that uses back-propagation, owing to the complexity of learning a series of sentences. These algorithms tend to generate redundant sentences, making generated dialogs inefficient [12, 13].

---

[*]Work carried out at Seoul National University

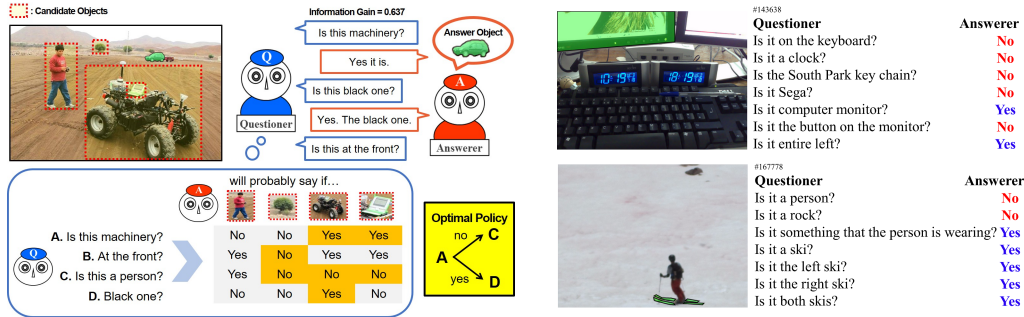

Figure 1: (Left) Illustration of an AQM algorithm for goal-oriented visual dialog. AQM makes a decision tree on the image for asking efficient questions. (Right) Examples of the GuessWhat?! game. The goal of GuessWhat?! is to locate the correct object in the image. The green mask highlights the correct object.

Our idea to deal with goal-oriented dialog is motivated by theory of mind [14], the ability to attribute mental states to others and to understand how our mental states are different. In this approach, an agent considers what the collaborator, the opposite agent cooperating in dialog, will respond by using an explicit approximated model of the collaborator. If one wishes to efficiently convey information to the other, it is best to converse in a way that maximizes the other's understanding [15]. For our method, we consider the mind to be beyond a part of mental states (e.g., belief, intent, knowledge). The mind is the probabilistic distribution of the model of the collaborator itself.

We propose an "Answerer in Questioner's Mind" (AQM) algorithm for goal-oriented dialog (Figure 1 (Left)). AQM allows a questioner to ask appropriate consecutive questions with information theoretic approach [16, 17]. AQM's questioner explicitly possesses an approximated model of the answerer. The questioner utilizes the approximated model to calculate the information gain of the candidate answerer's intentions and answers for each question. In the example of Figure 1 (Left), the answerer's intention is the correct object highlighted by the green mask. Using the candidate objects and the approximated answerer model, the questioner makes an efficient question which splits out the candidate objects properly. "Is this machinery?" is selected in the first turn, because the question separates the candidate objects evenly and thus has maximum information gain. In our main experiment, AQM's question generator extracts proper questions from the training data, not generating new questions. However, in the discussion section, we extend AQM to generate novel questions for the test image.

We test AQM mainly on goal-oriented visual dialog tasks in the self-play environment. Our main experiment is conducted on "GuessWhat?!", a cooperative two-player guessing game on the image (Figure 1 (Right)). AQM achieves an accuracy of 63.63% in 3 turns and 78.72% in 10 turns, outperforming deep SL (46.8% in 5 turns) [6] and deep RL (52.3% in 4.1 turns) [18] algorithms. Though we demonstrate the performance of our models in visual dialog tasks, our approach can be directly applied to general goal-oriented dialog where there is a non-visual context.

Our main contributions are four folds.

- We propose the AQM, a practical goal-oriented dialog system motivated by theory of mind. The AQM framework is general and not rely on a specific model representation nor a learning method. We compare various types of learning strategy on the model and selecting strategy for the candidate questions.

- We test our AQM on two goal-oriented visual dialog tasks, showing that our method outperforms comparative methods.

- We use AQM as a tool to understand existing deep learning methods in goal-oriented dialog studies. Section 5.1 and Appendix D include 1) the relationship between the hidden vector in comparative models and the posterior in AQM, 2) the relationship between the objective function of RL and AQM, and 3) a point to be considered on self-play with RL for making an agent to converse with a human.

- We extend AQM to generate questions, in which case AQM can be understood as a way to boost the existing deep learning method in Section 5.2.

## 2 Previous Works

Our study is related to various research fields, including goal-oriented dialog [1–3, 10, 11], language emergence [5, 19], the theory of mind [20–23], referring game [22, 24], pragmatics [25–28], and visual dialog [6, 7, 12, 13, 18]. In this section, we highlight three topics as below, obverter, opponent modeling, and information gain.

**Obverter** Choi et al. recently applied the obverter technique [21], motivated by theory of mind, to study language emergence [22]. The task of the study is an image-description-match classification. In their experiments, one agent transmitted one sentence for describing an artificial image to the collaborator agent. In their study, the obverter technique can be understood as that an agent plays both questioner and answerer, maximizing the consistency between visual and language modules. Their experimental results showed that their obverter technique generated a word corresponding to a specific object (e.g. 'bbbbbbb{b,d}' for a blue box). They argued their method could be an alternative to RL-based language emergence systems. Compared to their model, however, AQM uses real images, creates multi-turn dialog, and can be used for general goal-oriented dialog tasks.

**Opponent Modeling** Studies on opponent modeling have treated simple games with a multi-agent environment where an agent competed with the other [23]. In the study of Foerster et al. [29], the agent has the model of the opponent and updates it assuming the opponent will be updated by gradient descent with RL. They argued modeling opponent could be applied to track the non-stationary behavior of an opponent agent. Their model outperformed classical RL methods in simple games, such as tic-tac-toe and rock-paper-scissors. On the other hand, AQM applied opponent modeling to a cooperative multi-agent setting. We believe that opponent modeling can also be applied to dialog systems in which agents are partially cooperative and partially competitive.

In a broader sense, our study can also be understood as extending these studies to a multi-turn visual dialog, as the referring game is a special case of single-turn visual dialog.

**Information Gain** AQM's question-generator optimizes information gain using an approximated collaborator model. However, the concept of utilizing information gain in a dialog task is not new for a classical rule-based approach. Polifroni and Walker used information gain to build a dialog system for restaurant recommendations [30]. They made a decision tree using information gain and asked a series of informative questions about restaurant preferences. Rothe et al. applied a similar method to generate questions on a simple Battleship game experiment [31]. It is noticeable that they used pre-defined logic to generate questions with information gain criteria to make novel (i.e., not shown in the training dataset) and human-like questions. Unlike these previous studies, AQM makes a new decision tree for every new context; asking a question in AQM corresponds to constructing a node in decision tree classifier. In the example of Figure 1 (Left), AQM makes a new decision tree for a test image. AQM also considers uncertainty by deep learning, and does not require hand-made or domain-specific rules.

## 3 Answerer in Questioner's Mind (AQM)

**Preliminary** In our experimental setting, two machine players, a questioner and an answerer, communicate via natural dialog. Specifically, there exists a target class $c$, which is an answerer's intention or a goal-action the questioner should perform. The answerer knows the class $c$, whereas the questioner does not. The goal of the dialog for the questioner is to find the correct class $c$ by asking a series of questions to the answerer. The answerer responds the answer to the given question.

We treat $C$, $Q_t$, and $A_t$ as random variables of class, $t$-th question, and $t$-th answer, respectively. $c$, $q_t$, and $a_t$ becomes their single instance. In a restaurant scenario example, $q_t$ can be "Would you like to order?" or "What can I do for you?" $a_t$ can be "Two coffees, please." or "What's the password for Wi-Fi?" $c$ can then be "Receive the order of two hot Americanos." or "Let the customer know the Wi-Fi password."

**Model Structure** We illustrate the difference between the existing deep learning framework [6, 18, 13] and the proposed AQM framework in Figure 2. The answerer systems of two frameworks are the same. The answerer works as a kind of simple neural network models for visual question answering (VQA) and the network is trained on the training data. On the other hand, the questioner of two frameworks works differently.

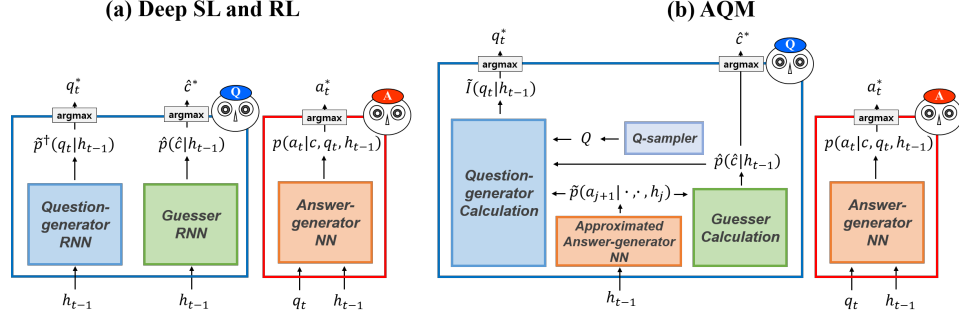

Figure 2: Comparative illustration on modules of the existing deep learning framework and an AQM framework. The AQM's guesser computes Equation 2 and the AQM's question-generator computes Equation 3. $h_{t-1}$ is the previous history $(a_{1:t-1}, q_{1:t-1})$.

In deep SL and RL methods [6, 18], the questioner has two RNN-based models, one is the question-generator to generate a question and the other is the guesser to classify a class. In AQM, these two RNN-based models are substituted by the mathematical calculation of the equation and argmax operation, not the model. The question-generator selects a question from candidate questions sampled from $Q$-sampler, which makes a pre-defined question set before the game starts. The question-generator calculates information gain $I[C, A_t; q_t, a_{1:t-1}, q_{1:t-1}]$ for each question in candidate questions. $I$ is the information gain or mutual information of the class $C$ and the current answer $A_t$, where the previous history $h_{t-1} = (a_{1:t-1}, q_{1:t-1})$ and the current question $q_t$ are given. Note that maximizing information gain is the same as minimizing the conditional entropy of class $C$, given a current answer $A_t$.

$$
\begin{aligned}
&I[C, A_t; q_t, a_{1:t-1}, q_{1:t-1}] \\
=&H[C; a_{1:t-1}, q_{1:t-1}] - H[C|A_t; q_t, a_{1:t-1}, q_{1:t-1}] \\
=&\sum_{a_t} \sum_c p(c|a_{1:t-1}, q_{1:t-1}) p(a_t|c, q_t, a_{1:t-1}, q_{1:t-1}) \ln \frac{p(a_t|c, q_t, a_{1:t-1}, q_{1:t-1})}{p(a_t|q_t, a_{1:t-1}, q_{1:t-1})}
\end{aligned}
\tag{1}
$$

The guesser calculates the posterior of class $p(c|a_{1:t}, q_{1:t}) \propto p(c) \prod_{j=1}^{t} p(a_j|c, q_j, a_{1:j-1}, q_{1:j-1})$.

**Calculation** For the equation of both question-generator and guesser, the answerer's answer distribution $p(a_t|c, q_t, a_{1:t-1}, q_{1:t-1})$ is required. The questioner has an approximated answer-generator network to make the approximated answer distribution $\tilde{p}(a_t|c, q_t, a_{1:t-1}, q_{1:t-1})$, which we refer to as the likelihood $\tilde{p}$. If $a_t$ is a sentence, the probability of the answer-generator is extracted from the multiplication of the word probability of RNN.

The AQM's guesser module calculates the posterior of class $c$, $\hat{p}(c|a_{1:t}, q_{1:t})$, based on the history $h_t = (a_{1:t}, q_{1:t})$, the likelihood $\tilde{p}$, and the prior of class $c$, $\hat{p}'(c)$. Using the likelihood model, the guesser selects a maximum a posterior solution for classifying the class.

$$
\hat{p}(c|a_{1:t}, q_{1:t}) \propto \hat{p}'(c) \prod_{j=1}^{t} \tilde{p}(a_j|c, q_j, a_{1:j-1}, q_{1:j-1})
\tag{2}
$$

We use a term of likelihood as $\tilde{p}$, prior as $\hat{p}'$, and posterior as $\hat{p}$ from the perspective that the questioner classifies class $c$. During the conversation, $\hat{p}$ can be calculated in a recursive way.

---

**Algorithm 1** AQM's Question-Generator

---

$\hat{p}(c) \sim \hat{p}'(c)$-model
$\tilde{p}(a_t|c, q_t, a_{1:t-1}, q_{1:t-1}) \sim \tilde{p}(a|c, q)$-model
$Q \leftarrow Q$-sampler
**for** $t = 1{:}T$ **do**
    $q_t \leftarrow \text{argmax}_{q_t \in Q} \ \tilde{I}[C, A_t; q_t, a_{1:t-1}, q_{1:t-1}]$ in Eq. 3
    Get $a_t$ from the answerer
    Update $\hat{p}(c|a_{1:t}, q_{1:t}) \propto \tilde{p}(a_t|c, q_t, a_{1:t-1}, q_{1:t-1}) \cdot \hat{p}(c|a_{1:t-1}, q_{1:t-1})$
**end for**

---

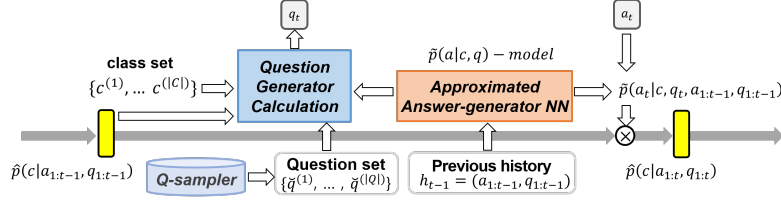

Figure 3: Procedural illustration on the AQM's question-generator.

The AQM's question-generator module selects $q_t^*$, which has a maximum value of the approximated information gain $\tilde{I}[C, A_t; q_t, a_{1:t-1}, q_{1:t-1}]$, or simply $\tilde{I}$. To calculate the information gain $\tilde{I}$, the question-generator module uses the likelihood $\tilde{p}$ and the posterior $\hat{p}$.

$$
\begin{aligned}
q_t^* &= \underset{q_t \in Q}{\text{argmax}} \ \tilde{I}[C, A_t; q_t, a_{1:t-1}, q_{1:t-1}] \\
&= \underset{q_t \in Q}{\text{argmax}} \sum_{a_t} \sum_{c} \hat{p}(c|a_{1:t-1}, q_{1:t-1}) \tilde{p}(a_t|c, q_t, a_{1:t-1}, q_{1:t-1}) \ln \frac{\tilde{p}(a_t|c, q_t, a_{1:t-1}, q_{1:t-1})}{\tilde{p}'(a_t|q_t, a_{1:t-1}, q_{1:t-1})}
\end{aligned}
\tag{3}
$$

where $\tilde{p}'(a_t|q_t, a_{1:t-1}, q_{1:t-1}) = \sum_c \hat{p}(c|a_{1:t-1}, q_{1:t-1}) \cdot \tilde{p}(a_t|c, q_t, a_{1:t-1}, q_{1:t-1})$. $Q$-sampler is required to select the question from the candidate questions $Q$. In our main experiments in Section 4.2, $Q$-sampler extracts candidate questions from the training data. In this case, AQM does not generate a new question for the test image. However, if $Q$-sampler uses a RNN-based model, AQM can generate the question. We discuss this issue in Section 5.2.

**Learning** In AQM, the answer-generator network in the questioner and the answerer does not share the representation. Thus, we need to train the AQM's questioner. In the existing deep learning framework, SL and RL are used to train two RNN-based models of the questioner. In a typical deep SL method, questioner's RNNs are trained from the training data, which is the same or similar to the data the answerer is trained from. In a typical deep RL method, the answerer and the questioner make a conversation in the self-play environment. In this RL procedure, the questioner uses the answers generated from the answerer for end-to-end training, with reward from the result of the game. On the other hand, the AQM's questioner trains the approximated answer distribution of the answerer, the likelihood $\tilde{p}$. The likelihood $\tilde{p}$ can be obtained by learning training data as in deep SL methods, or using the answers of the answerer obtained in the training phase of the self-play conversation as in deep RL methods. As the objective function of RL or AQM does not guarantee human-like question generation [32], RL uses SL-based pre-training, whereas AQM uses an appropriate $Q$-sampler.

Algorithm 1 and Figure 3 explain the question-generator module procedure. The question-generator requires the $\hat{p}'(c)$-model for the prior, the $\tilde{p}(a|c, q)$-model for the likelihood, and the $Q$-sampler for the set of candidate questions. Additional explanations on AQM can be found in Appendix A.

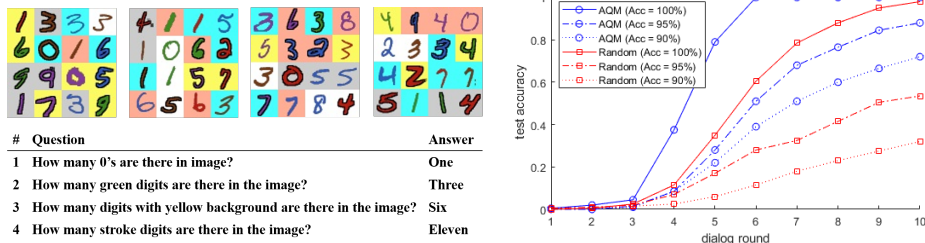

| # | Question | Answer |
|---|---|---|
| 1 | How many 0's are there in image? | One |
| 2 | How many green digits are there in the image? | Three |
| 3 | How many digits with yellow background are there in the image? | Six |
| 4 | How many stroke digits are there in the image? | Eleven |

Figure 4: (Left) Illustration of MNIST Counting Dialog, a simplified version of MNIST Dialog [33]. (Right) Test accuracy of goal-oriented dialog from the MNIST Counting Dialog task. "Acc" is PropAcc, the average ratio of property recognition accuracy.

## 4  Experiments

### 4.1  MNIST Counting Dialog

To clearly explain the mechanism of AQM, we introduce the MNIST Counting Dialog task, which is a toy goal-oriented visual dialog problem, illustrated in Figure 4 (Left). Each image in MNIST Counting Dialog contains 16 small images of digit, each having four randomly assigned properties: color = {red, blue, green, purple, brown}, bgcolor = {cyan, yellow, white, silver, salmon}, number = {0, 1, · · · , 9}, and style = {flat,stroke}. The goal of the MNIST Counting Dialog task is to inform the questioner to pick the correct image among 10K candidate images via questioning and answering. In other words, class $c$ is an index of the true target image (1 ∼ 10,000).

For the MNIST Counting Dialog task, we do not model the questioner and the answerer using neural networks. Instead, we define the answer model in the questioner is count-based on 30K training data. We set the average ratio of property recognition accuracy (PropAcc) $\lambda$ as 0.9, 0.95, and 1.0 in Figure 4. For four properties such as color, bgcolor, number and style, each property recognition accuracy $\lambda_{color}$, $\lambda_{bgcolor}$, $\lambda_{number}$, $\lambda_{style}$ is randomly sampled from an uniform distribution with a range of $[(2\lambda - 1), 1]$. It contributes to add randomness or uncertainty to this task. For example, the percentage of correctly recognized color is 88% if $\lambda_{color}$ is 0.88.

According to the results in Figure 4, if the PropAcc $\lambda$ decreases, the accuracy of goal-oriented dialogue which has a goal to select the correct image is decreased by a large amount. Figure 4 (Right) shows that AQM nearly always chooses the true target image from 10K candidates in six turns if the PropAcc $\lambda$ is 1.0. However, AQM also chooses correctly with a probability of 51% and 39% (accuracy of goal-oriented dialog) in six turns, when the PropAcc $\lambda$ is 0.95 and 0.90, respectively. "Random" denotes a questioner with a random question-generator module and the AQM's guesser module. Detailed experimental settings can be found in Appendix B.

### 4.2  GuessWhat?!

**GuessWhat?! Task**  GuessWhat?! is a cooperative two-player guessing game proposed by De Vries et al. (Figure 1 (Right)) [6]. GuessWhat?! has received attention in the field of deep learning and artificial intelligence as a testbed for research on the interplay of computer vision and dialog systems. The goal of GuessWhat?! is to locate the correct object in a rich image scene by asking a sequence of questions. One participant, "Answerer", is randomly assigned an object in the image. The other participant, "Questioner," guesses the object assigned to the answerer. Both a questioner and an answerer sees the image, but the correct object is known only to the answerer. To achieve the goal, the questioner asks a series of questions, for which the answerer responds as "yes," "no," or "n/a." The questioner does not know a list of candidate objects while asking questions. When the questioner decides to guess the correct object, a list of candidate objects is then revealed. A win occurs when the questioner picks the correct object. The GuessWhat?! dataset contains 66,537 MSCOCO images [34], 155,280 games, and 831,889 question-answer pairs.

$\hat{p}'(c)$-**model for the Prior**  The questioner does not know the list of candidate objects while asking questions. This makes the GuessWhat?! task difficult, although the number of candidates is around 8.

Figure 5: Test accuracy from the GuessWhat?!. Previous works do not report the performance change with an increase in the number of turns.

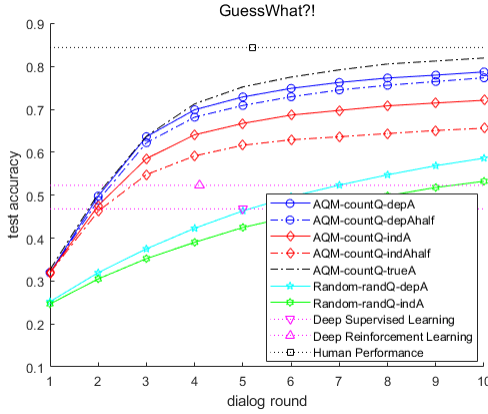

Table 1: Test accuracy from the GuessWhat?!.

| Model | Accuracy |
|---|---|
| Baseline | 16.04 |
| Deep SL (5-q) [6] | 46.8 |
| Deep RL (4.1-q in Avg) [18] | 52.3 |
| **Random-randQ-indA (5-q)** | 42.48 ($\pm$ 0.84) |
| **Random-randQ-depA (5-q)** | 46.36 ($\pm$ 0.91) |
| **AQM-randQ-indA (5-q)** | 65.66 ($\pm$ 0.55) |
| **AQM-countQ-indAhalf (5-q)** | 61.64 ($\pm$ 0.97) |
| **AQM-countQ-indA (5-q)** | 66.73 ($\pm$ 0.76) |
| **AQM-countQ-depAhalf (5-q)** | 70.90 ($\pm$ 1.14) |
| **AQM-countQ-depA (5-q)** | 72.89 ($\pm$ 0.70) |
| **AQM-countQ-depAhalf (10-q)** | 77.35 ($\pm$ 0.85) |
| **AQM-countQ-depA (10-q)** | **78.72** ($\pm$ 0.54) |
| Human [18] | 84.4 |

We use YOLO9000, a real-time object detection algorithm, to estimate the set of candidate objects [35]. The prior $\hat{p}'(c)$ is set to $1/N$, where $N$ is the number of extracted objects.

$\tilde{p}(a|q, c)$**-model for the Likelihood** We use the answerer model from previous GuessWhat?! research [6]. The inputs of the answer-generator module consist of a VGG16 feature of a given context image, a VGG16 feature of the cropped object in the context image, spatial and categorical information of the cropped object, and the question $q_t$ at time step $t$. A simple multi-layer perceptron model uses these features to classify the answer {yes, no, n/a}. Our answer-generator module assumes the answer distribution is independent from the history $h_{t-1} = (a_{1:t-1}, q_{1:t-1})$. In other words, we approximate the likelihood as $\tilde{p}(a_t|c, q_t, a_{1:t-1}, q_{1:t-1}) \propto \tilde{p}''(a_t|c, q_t)$.

We use various strategy to train the questioner's approximated answer-generator network $\tilde{p}(a_t|c, q_t, a_{1:t-1}, q_{1:t-1})$ to approximate the answerer's answer distribution $p(a_t|c, q_t, a_{1:t-1}, q_{1:t-1})$. In "indA," $p$ and $\tilde{p}$ is trained separately for the same training data. In "depA," in which $\tilde{p}$ is trained for the answer inferred from the answerer $p$, where the question and the image is also sampled from the training data. The performance improvement of indA and depA setting would be achieved partly because the answerer and the questioner share the dataset. For ablation study, we also introduce "indAhalf" and "depAhalf" setting. In indAhalf, $p$ and $\tilde{p}$ is trained for the different dataset each, which the training data is exclusively divided into halves. In depAhalf, $p$ is trained for the first half training data. After that, the questioner asks the questions in the second half training data to get the answer from $p$ and use the answer as the training label.

$Q$**-sampler for the Candidate Question Set** In the main experiments, we compare two $Q$-samplers which select the question from the training data. The first is "randQ," which samples questions randomly from the training data. The second is "countQ," which causes every other question from the set $Q$ to be less dependent on the other. countQ checks the dependency of two questions with the following rule: the probability of that two sampled questions' answers are the same cannot exceed 95%. In other words, $\sum_a \tilde{p}^\dagger(a_i = a|q_i, a_j = a, q_j) < 0.95$, where $\tilde{p}^\dagger(a_i|q_i, a_j, q_j)$ is derived from the count of a pair of answers for two questions in the training data. $\tilde{p}$ made by indA is used for countQ. We set the size of $Q$ to 200.

**Experimental Results** Figure 5 and Table 1 shows the experimental results. Figure 6 illustrates the generated dialog. Our best algorithm, AQM-countQ-depA, achieved 63.63% in three turns, outperforming deep SL and deep RL algorithms. By allowing ten questions, the algorithms achieved 78.72% and reached near-human performance. If the answerer's answer distribution $p$ is directly used for the questioner's answer distribution $\tilde{p}$ (i.e., $\tilde{p} = p$), AQM-countQ achieved 63.76% in three turns and 81.96% in ten turns (AQM-countQ-trueA in Figure 5). depA remarkably improved the score in self-play but did not increased the quality of the generated dialog significantly. On the other hand, countQ did not improve the score much but increased the quality of the generated dialog. It is noticeable that the performance gap between indAhalf and indA is much larger than the gap

| | Groundtruth | Supervised Learning (SL) | Answerer in Questioner's Mind (AQM, countQ) | Answerer in Questioner's Mind (AQM, gen1Q) |
|---|---|---|---|---|
| 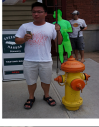 | Is it person? Is he in the foreground? Is he wearing blue? | Is it a person? Is it the one in the front? Is it the one in the middle? Is it the one in the middle? Is it the whole person? | Is it a person? Is it a player? Is it in the background? Is it the umpire? Single piece in the bunch? | Is it one of the people? Is it in the background? Is it the person on the right? Is it the person holding the cake? Is it the person or the fire hydrant? |
|  | Is it an animal? Is it a device? Is it silver in color? | Is it the cat? Is it the cat? Is it the chair? Is it the book? Is it the book? | Is it alive? A laptop? Is it the shelf portion on the right? Left side of green color? Is a table? | Is it on the left half of the screen? Is it a book? Is it one of the bottles? Is it one of the cats? Is it on the table? |

| | Groundtruth | Reinforcement Learning (RL) | Answerer in Questioner's Mind (AQM, countQ) | Answerer in Questioner's Mind (AQM, gen1Q) |
|---|---|---|---|---|
|  | Is it a person? | Is it a person? Is it girl in white? | Is it a fork? Is it a chair? Is it a food? Is it a person? Is it alive? | Is it on the table? Is the object human? Is it human? Is it one of the people? Is it a human? |
|  | Is a person? Is it tree? Is an object? The bat playing? The shoes on food? Is the red seat? | Is it a person? Is it a love? Is an umbrella? Is in the middle? On a person? Is it on on far right? | Is it a person? Is it the chair? Is it a traffic signal light? Is it the phone? Is it alive? | Is it human? Is it the man or part of the ball? Is it the person on the right? Is it a person or something they are holding? Is it being held by a person? |

Figure 6: Generated dialogs from our algorithm and the comparative algorithms. The tested games are sampled from the selected results of previous papers [6, 18].

between depAhalf and depA. This result shows that the conversation between the questioner and the answerer affects AQM's performance improvement more than sharing the training data between the questioner and the answerer. The compared deep SL method used the question-generator with the hierarchical recurrent encoder-decoder [36], achieving an accuracy of 46.8% in five turns [6]. However, Random-randQ-depA achieved 46.36% in five turns, which is a competitive result to the deep SL model. "Random" denotes random question generation from the randQ set. The comparative deep RL method applied reinforcement learning on long short-term memory, achieving 52.3% in about 4.1 turns [18]. The deep RL has a module to decide whether the dialog is stopped or not. 4.1 is the number of the averaged turns.

# 5 Discussion

## 5.1 Comparing AQM with SL and RL

Our study replaced the task of training an RNN which generates questions with the task of training a neural network which infers the probability of the answers. In the perspective of the hidden representation to track the dialog, the contexts of history which AQM's questioner requires are the posterior $\hat{p}$ and the history itself $h_t = (a_{1:t}, q_{1:t})$ used as an input for the likelihood $\tilde{p}$. In deep SL and RL methods, hidden neurons in RNN are expected to track the context of history. If the question to be asked is independent from the previous questions, the only context AQM should track is the posterior $\hat{p}$. In this case, the posterior $\hat{p}$ in the yellow box of Figure 3 corresponds to the hidden vector of the RNN in the comparative dialog studies.

Moreover, we argue that, in Appendix D, AQM and RL have a similar objective function, just as information gain in decision tree is used to classify. Many researchers have studied dialog systems for cooperative games using deep RL, to increase the score in self-play environments [32, 37]. In Appendix D, we leverage AQM as a tool for analyzing the deep RL approach on goal-oriented dialog tasks from the perspective of theory of mind. According to our argument, training two agents to make plausible dialogs via rewards during self-play is not adaptable to a service scenario. To enable an agent to converse with a human, the opponent agent in self-play should model a human as much as possible. We prove that AQM and RL have a similar objective function, implying that RL-based training for the questioner can also be seen as implicit approximation on the answer distribution of the answerer.

## 5.2 Generating Questions

For making a question more relevant to the new context of the dialog, a question needs to be generated. Extracting a question from the training dataset (randQ and countQ) is just one of the settings the AQM's $Q$-sampler can have. AQM can generate questions by using rule-based program [31] or a seq2seq models previously used in goal-oriented dialog studies. As a simple extension, we test the "gen1Q" setting, which uses a previous deep SL question-generator [6] to generate a first-turn question for the test image. We use a beam search to make a set of questions sorted by its likelihood, and select top-100 questions for the candidates. In the experiments on GuessWhat?!, AQM-gen1Q-depA achieves a slight performance improvement over AQM-countQ-depA at 2-q (49.79% $\rightarrow$ 51.07%) outperforming the original deep SL method (46.8% in 5-q). However, at 5-q, AQM-gen1Q performs slightly worse than AQM-countQ-depA (72.89% $\rightarrow$ 70.74%). If the Q-sampler generates questions through the seq2seq model using the history of dialog at every turn, the performance would be improved further. Appendix E discusses the further direction of future works to make the question agent applicable for service.

Figure 6 shows the generated dialogs of AQM-gen1Q. gen1Q tends to make more redundant sentences than countQ because countQ carefully checks the similarity between questions. However, gen1Q tends to make the questions more related to the image. It is also noticeable that there are questions concatenating two sentences with "or" in gen1Q. The score of the game is insufficient to evaluate the quality of the generated dialog. Appendix C discusses the objective function of the goal-oriented dialog, mainly based on the case of goal-oriented visual dialog studies.

## 6 Conclusion

We proposed "Answerer in Questioner's Mind" (AQM), a practical goal-oriented dialog framework using information-theoretic approach. In AQM, the questioner approximates the answerer's answer distribution in dialog. In our experiments, AQM outperformed deep SL and RL methods. AQM can be implemented in various manners, not relying on a specific model representation nor a learning method. We also extended AQM to generate question by applying a previously proposed deep SL method. In this case, AQM can be understood as a way to boost the existing deep learning method. Throughout the paper, we argued that considering the collaborator's mind in implementing an agent is useful and fundamental.

## Acknowledgements

The authors would like to thank Jin-Hwa Kim, Tong Gao, Cheolho Han, Wooyoung Kang, Jaehyun Jun, Hwiyeol Jo, Byoung-Hee Kim, Kyoung Woon On, Sungjae Cho, Joonho Kim, Seungjae Jung, Hanock Kwak, Donghyun Kwak, Christina Baek, Minjoon Seo, Marco Baroni, and Jung-Woo Ha for helpful comments and editing. This work was supported by the Institute for Information & Communications Technology Promotion (R0126-16-1072-SW.StarLab, 2017-0-01772-VTT , 2018-0-00622-RMI) and Korea Evaluation Institute of Industrial Technology (10060086-RISF) grant funded by the Korea government (MSIP, DAPA).

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
