[Supplementary Material]

## Appendix A. Additional Explanations on AQM

In our problem setting, three modules exist: an answer-generator, a question-generator and a guesser. The objective function of three modules is as follows.

Answer-generator: $\mathrm{argmax}_{a_t}\, p(a_t|c, q_t, a_{1:t-1}, q_{1:t-1})$

Guesser: $\mathrm{argmax}_c\, p(c|a_{1:t}, q_{1:t})$

Question-generator: $\mathrm{argmax}_{q_t}\, I[C, A_t; q_t, a_{1:t-1}, q_{1:t-1}]$

If $a_t$ is a sentence, the probability of the answer-generator is extracted from the multiplication of the word probability of RNN. For the objective function of question-generator, we use information gain $I$ of the class $C$ and the current answer $A_t$, where the previous history $(a_{1:t-1}, q_{1:t-1})$ and a current question $q_t$ are given.

$$
\begin{aligned}
&I[C, A_t; q_t, a_{1:t-1}, q_{1:t-1}] \\
=&H[C; a_{1:t-1}, q_{1:t-1}] - H[C|A_t; q_t, a_{1:t-1}, q_{1:t-1}] \\
=& -\sum_c p(c|a_{1:t-1}, q_{1:t-1}) \ln p(c|a_{1:t-1}, q_{1:t-1}) \\
& + \sum_{a_t}\sum_c p(c, a_t|q_t, a_{1:t-1}, q_{1:t-1}) \ln p(c|a_t, q_t, a_{1:t-1}, q_{1:t-1}) \\
=& \sum_{a_t}\sum_c p(c, a_t|q_t, a_{1:t-1}, q_{1:t-1}) \ln \frac{p(c|a_t, q_t, a_{1:t-1}, q_{1:t-1})}{p(c|a_{1:t-1}, q_{1:t-1})} \\
=& \sum_{a_t}\sum_c p(c|a_{1:t-1}, q_{1:t-1}) p(a_t|c, q_t, a_{1:t-1}, q_{1:t-1}) \ln \frac{p(a_t|c, q_t, a_{1:t-1}, q_{1:t-1})}{p(a_t|q_t, a_{1:t-1}, q_{1:t-1})}
\end{aligned}
\tag{4}
$$

For the derivation of the last line, Bayes rule is used. We also assume that $p(c|q_t, a_{1:t-1}, q_{1:t-1}) = p(c|a_{1:t-1}, q_{1:t-1})$, because asking a question itself does not affect the posterior of class $c$.

In calculating information gain $I$, AQM directly assigns each probability value to the equation of $I$, and sums out $a_t$ and $c$. However, all probability functions in Equation 4 are the distribution from the answerer. Our approach is to approximate the answerer's answer distribution $p(a_t|c, q_t, a_{1:t-1}, q_{1:t-1})$ to the likelihood $\tilde{p}(a_t|c, q_t, a_{1:t-1}, q_{1:t-1})$. We use the approximated information gain $\tilde{I}$ to select an adequate question.

$$
\begin{aligned}
&\tilde{I}[C, A_t; q_t, a_{1:t-1}, q_{1:t-1}] \\
=&\sum_{a_t}\sum_c \hat{p}(c|a_{1:t-1}, q_{1:t-1})\tilde{p}(a_t|c, q_t, a_{1:t-1}, q_{1:t-1}) \ln \frac{\tilde{p}(a_t|c, q_t, a_{1:t-1}, q_{1:t-1})}{\tilde{p}'(a_t|q_t, a_{1:t-1}, q_{1:t-1})}
\end{aligned}
\tag{5}
$$

In calculating $\tilde{I}$, $\tilde{p}$ can be obtained by approximated answer distribution model. However, there are other probability functions $\hat{p}$, $\hat{p}'$, and $\tilde{p}'$ beside the answer distribution. These three functions can be again calculated from $\tilde{p}$. The posterior $\hat{p}$ can be calculated by the likeilhood $\tilde{p}$ and the prior $\hat{p}'$.

$$
\hat{p}(c|a_{1:t}, q_{1:t}) \propto \hat{p}'(c) \prod_{j=1}^{t} \tilde{p}(a_j|c, q_j, a_{1:j-1}, q_{1:j-1})
\tag{6}
$$

During the conversation, the posterior can be calculated in a recursive way using $\hat{p}(c|a_{1:t}, q_{1:t}) \propto \tilde{p}(a_t|c, q_t, a_{1:t-1}, q_{1:t-1}) \cdot \hat{p}(c|a_{1:t-1}, q_{1:t-1})$. $\hat{p}'$ needs to be pre-defined. It can be a uniform distribution or other knowledge can be also applied to give adequate prior. $\tilde{p}'$ can be again calculated with $\tilde{p}$ and $\hat{p}$.

$$
\tilde{p}'(a_t|q_t, a_{1:t-1}, q_{1:t-1}) = \sum_c \hat{p}(c|a_{1:t-1}, q_{1:t-1})\tilde{p}(a_t|c, q_t, a_{1:t-1}, q_{1:t-1})
\tag{7}
$$

When the answerer's answer distribution $p(a_t|c, q_t, a_{1:t-1}, q_{1:t-1})$ is fixed, the questioner achieves an ideal performance when the likelihood $\tilde{p}$ is the same as $p(a_t|c, q_t, a_{1:t-1}, q_{1:t-1})$.

## Appendix B. MNIST Counting Dialog

To clearly explain the mechanism of AQM, we introduce the MNIST Counting Dialog task, which is a toy goal-oriented visual dialog problem, illustrated in Figure 4 (Left). In the example of Figure 4 (Left), asking about the number of 1 digits or 6 digits classifies a target image perfectly if property recognition accuracy on each digit in the image $\lambda_{number}$ is 100%. Asking about the number of 0 digits does not help classify, because all images have one zero. If $\lambda_{number}$ is less than 100%, asking about the number of 1 digits is better than asking about the number of 6 digits, because the variance of the number of 1 digits is larger than that of the 6 digits.

An answering model in questioner is trained for 30K training data. 22 questions (for from red to stroke) and corresponding 22 answerer's answers are used for learning each instance of the training data. Answering model in questioner is count-based. For true answer $a^{real}$ and answerer's answer $a^{feat}$ of the question $q_t$, the questioner's likelihood $\tilde{p}$ is as follow.

$$\tilde{p}(a_t|c, q_t, a_{1:t-1}, q_{1:t-1}) \propto \tilde{p}''(a_t|c, q_t)$$
$$= \frac{\#a^{feat}|a^{real} + \epsilon}{\#a^{real} + \epsilon'} \tag{8}$$

This equation uses an independence assumption. $\#a^{real}$ is the number of $a^{real}$ for the training data where the answer for the class $c$ and the question $q_t$ is $a^{real}$. $\#a^{feat}|a^{real}$ is the number of case where $a^{real}$ and $a^{feat}$ appears together. $\epsilon$ and $\epsilon'$ is the constant for normalization. As the questioner uses the answer inferred from the answerer in the training phase, our setting in MNIST Counting Dialog corresponds to depA in GuessWhat?!.

## Appendix C. Objective Function of Goal-Oriented Visual Dialog

There have been several kinds of visual-language tasks including image captioning [38] and VQA [39, 24], and recent research goes further to propose multi-turn visual dialog tasks [12]. This section discusses the objective functions in research on goal-oriented dialog, mainly based on the case of goal-oriented visual dialog studies [6, 13].

**Visual Dialog**   In Visual Dialog [7], two agents also communicate with questioning and answering about the given MSCOCO image. Unlike GuessWhat?!, an answer can be a sentence and there is no restriction for the dialog answerer. Das et al. used this dataset to make a goal-oriented dialog task, where the questioner guesses a target image from 9,627 candidates in the test dataset [13]. It is noticeable that, however, the questioner only uses the caption of the image, not the image itself when generating questions, because the goal of the task is to figure out the target image. The dataset includes a true caption of each image achieving percentile ranks of around 90%. In their self-play experiments, adding information via a dialog improved the percentile ranks to around 93%, where the questioner and answerer were trained with deep SL and deep RL methods. This means that the models predict the correct image to be more exact than 93% of the rest images in the test dataset. Note that their models only improved around 3% of the percentile rank at most, which implies that the caption information is more important than the dialog. There are also relevant issues on the performance which can be informed in the author's Github repository[2]. According to their explanation, 0-th turn the percentile rank (only using the caption information) could be improved further by fine-tuning the hyperparameters, but in this case, the percentile rank did not increase much (around 0.3 of the percentile rank) during the conversation.

**Objective 1: Score**   In goal-oriented visual dialog research, the score is used as one of the main measurements of dialog efficiency. However, a high score can be achieved via optimization over $p(c|a_{1:T}, q_{1:T})$, which is the objective function of RL. In particular, the agent can achieve a high score although the probability distribution of both the questioner and answerer is not bound to a human's distribution, even when human-like dialog is generated. For example, in GuessWhat?!, Han

et al. showed that pre-defined questions about location can provide an accuracy of 94.34% in five turns [40]. Their methods divided an image evenly into three parts using two vertical or horizontal lines for three cases of answer {yes, no, n/a}. A natural language-based protocol can also be created using size, color, category, or other major properties of the object.

**Objective 2: Service** One of the ultimate objectives of goal-oriented dialog research is to create an agent that can be used in a real service [3]. However, successful reports have been limited. Chattopadhyay et al. reported the human-machine performance of the deep RL method in a study on Visual Dialog [13, 41]. In their method, questioner and answerer were both fine-tuned by RL. Thus, the answerer's answering distribution differed from the training data. This RL method also used the objective function of deep SL as the regularizer, conserving generated dialog as human-like. Nevertheless, the RL algorithm deteriorated the score in the game with a human, compared to deep SL. The authors also assessed six measures of generated dialog quality: accuracy, consistency, image understanding, detail, question understanding, and fluency. However, the human subjects reported that the deep RL algorithm performed worse than the deep SL algorithm for all measures, except "detail."

**Objective 3: Language Emergence** Plenty of research has recently been published on language emergence with RL in a multi-agent environment. Some studied artificial (i.e., non-natural) language [19, 5], whereas others attempted to improve the quality of generated natural dialog. One of the best progress found in the latter study is the improvement on the quality of a series of questions in a multi-turn VQA. When deep RL is applied, the questioner generates fewer redundant questions than deep SL [18, 13]. It can be understood that the questioner in these methods are optimized by both $p(q_t|a_{1:t-1}, q_{1:t-1})$ and $p(c|a_{1:T}, q_{1:T})$. Deep RL methods use deep SL algorithms as a pre-training method [18], or use the objective function of $p(c|a_{1:T}, q_{1:T}) + \lambda \cdot p(q_t|a_{1:t-1}, q_{1:t-1})$ [13]. These studies focused on or achieved improvement of the questioner more than the answerer. It is because the answerer gets the objective function directly for each answer (i.e., VQA), whereas the questioner does not.

## Appendix D. Analyzing RL via AQM

`AQM's Property 1.` *The performance of AQM's questioner is optimal, where the likelihood $\tilde{p}$ is equivalent to the answering distribution of the opponent $p$.* For the guesser module, the performance of the guesser with the posterior $\hat{p}$ is optimal when $\tilde{p}$ is $p$. The performance of question-generator also increases as the $\tilde{p}$ becomes more similar to the opponent, when the $\hat{p}$ is fixed.

`RL's Property 1.` *AQM and RL approaches share the same objective function.* Two algorithms have the same objective function with the assumption that $q_t$ only considers the performance of the current turn. The assumption is used in the second line of the following equation.

$$
\begin{aligned}
& \underset{q_t}{\arg\max} \ \underset{q_{t-}}{\max} \ln p(c|a_{1:T}, q_{1:T}) \\
\approx \ & \underset{q_t}{\arg\max} \ \ln p(c|a_{1:t}, q_{1:t}) \\
= \ & \underset{q_t}{\arg\max} \ E\left[\ln \frac{p(a_t|c, q_t, a_{1:t-1}, q_{1:t-1})}{p(a_t|q_t, a_{1:t-1}, q_{1:t-1})}\right] \\
= \ & \underset{q_t}{\arg\max} \ I[C, A_t; q_t, a_{1:t-1}, q_{1:t-1}]
\end{aligned}
\tag{9}
$$

$q_{t-}$ denotes $\{q_{1:t-1}, q_{t+1:T}\}$. In the third line, $a_t \sim p(a_t|c, q_t, a_{1:t-1}, q_{1:t-1})$, $c \sim p(c|a_{1:t-1}, q_{1:t-1})$, and Bayes rule is used. The assumption in the second line can be alleviated via multi-step AQM, which uses $I[C, A_{t:t+k}; q_{t:t+k}, a_{1:t-1}, q_{1:t-1}]$ as the objective function of the question-generator module. In the multi-step AQM, the optimal question cannot be selected in a greedy way, unlike original AQM. The multi-step AQM needs to search in a tree structure; a Monte Carlo tree search [42] can be used to find a reasonable solution.

The RL and AQM question-generator are closely related, as is the discriminative-generative pair of classifiers [43]. AQM's question-generator and guesser module explicitly have a likelihood $\tilde{p}$, whereas the RL's modules do not have explicitly. The properties of AQM including optimal conditions and sentence dynamics can be extended to RL. The complexity of RL's question-generator can be

decomposed to tracking class posteriors $p(c|a_{1:t-1}, q_{1:t-1})$ and history $(a_{1:t-1}, q_{1:t-1})$ for multi-turn question answering. For human-like learning, the context for language generation $p(q_t|a_{1:t-1}, q_{1:t-1})$ is also required; $Q$-sampler corresponds to this context.

`RL's Property 2.` *Optimizing both questioner and answerer with rewards makes the agent's performance with human worse.* This is true, even when the process improves a score during self-play or uses tricks to maintain a human-like language generation. The property of self-play in a cooperative goal-oriented dialog task is different from the case of AlphaGo, which defeated a human Go champion [44]. For example, in GuessWhat?!, the reversed response of the answerer (e.g., "no" for "yes") may preserve the score in the self-play, but it makes the score in the human-machine game near 0%.

According to AQM's Property 1, for the play of a machine questioner and a human answerer, the performance is optimal only if the approximated answerer's distribution $\tilde{p}$ of AQM's questioner is the same as the human's answering distribution. According to RL's Property 1, RL and AQM shares the optimality condition about the distribution of the opponent. Fine-tuning an answerer agent with a reward makes the distribution of the agent different from a human's. Therefore, fine-tuning both agents decreases performance in service situations. The experiment with a human, studied by Chattopadhyay et al. explained in Appendix C, empirically demonstrates our claim.

`RL's Property 3.` *An alternative objective function exists, which is directly applicable to each question.* A reward can only be applied when one game is finished. If the goal of training is to make language emergence itself or to make an agent for service, two agents can communicate with more than just question-answering for back-propagation, such as sharing attention for the image [12]. Cross-entropy for the guesser of each round can be considered to replace the reward. Information gain can also be used not only for AQM but also for alternative objectives of back-propagation.

## Appendix E. Future Works

**RL with Theory of Mind** RL methods can be enhanced in a service scenario by considering the answering distribution of human. It is advantageous for the machine questioner to ask questions for which the human answer is predictable [20]. In other words, a question having a high VQA accuracy is preferred. The model uncertainty of the questioner can also be measured and utilized with recent studies on Bayesian neural networks and uncertainty measures [45]. Because the questioner has the initiative of dialog, the questioner does not need to necessarily learn the entire distribution of human conservation. The question, which the questioner uses frequently in self-play, can be asked more to a human. Then, the obtained question-answer pairs can be used for improving the answerer, like in active learning.

**Combining Seq2seq with AQM** RL optimizes $p(c|a_{1:t}, q_{1:t})$ for the questioner in goal-oriented dialog. In the perspective of natural dialog generation, however, RL can be understood as that the questioner are optimized by both $p(q_t|a_{1:t-1}, q_{1:t-1})$ and $p(c|a_{1:T}, q_{1:T})$, as described in Appendix D. On the other hand, the $Q$-sampler in AQM corresponds to regularizing with $p(q_t|a_{1:t-1}, q_{1:t-1})$ in the deep RL approach. If the $Q$-sampler in our experiment is replaced with seq2seq trained by a deep SL method, AQM can generate a question by optimizing both sentence probability from the seq2seq model and information gain. AQM can use following terms in every turn to generate questions.

$$I[C, A_t; q_t, a_{1:t-1}, q_{1:t-1}] + \lambda \cdot \tilde{p}^{\dagger}(q_t|a_{1:t}, q_{1:t}) \tag{10}$$

$\lambda$ is a balancing parameter. $\tilde{p}^{\dagger}$ is a probability distribution of language modeling of the seq2seq model. Comparing with the gen1Q setting in the discussion section, this algorithm could make the dialog more related to the history of dialog. In this idea, AQM can be understood as a way to boost deep SL methods.

**Online Learning** Fine-tuning on the model is required for a novel answerer, a non-stationary environment [29], or a multi-domain problem. We think that fine-tuning on the answerer model would be more robust than on the question-generating RNN model, making AQM would have an advantage, because research on training an answerer model on VQA tasks has been more progressed than training an RNN for the questioner. On the other hand, if the experiences of many users are available, model-agnostic meta learning (MAML) can be applied for few-shot learning [46].

## Footnotes

[2]https://github.com/batra-mlp-lab/visdial-rl