[Reviews · NeurIPS 2018]

Reviewer 1



— This paper introduces an approach to approximate the answerer within the questioner for the GuessWhat?! game involving goal-oriented visually-grounded dialog between a questioner + guesser and answerer. — The proposed questioner has 3 components — the question-sampler (proposing candidate questions), the approximated answerer (predicting answers for each candidate question; and pretrained on the training set and/or via imitation of the actual answerer), object class guesser, and the question-picking calculation module. — The question-picking module calculates mutual information of the object class and answer given history and a candidate question, and the argmax informative question is picked. — Experiments are performed on MNIST counting dialog and GuessWhat. For MNIST, the approximated answerer is count-based and its recognition accuracy can be controlled proportional to the actual answerer’s accuracy. For GuessWhat, the approximated answerer is trained in a variety of ways — on the same training data as the actual answerer, on predicted answers from the actual answerer, on a different training data split as the actual answerer, and on a different training data split as the actual answerer followed by imitation of predicted answers on the other split. — On MNIST, the authors compare against a random question-picker module. And the proposed approach outperforms the random baseline. — On GuessWhat, the proposed approach outperforms prior work using supervised and reinforcement learning without an explicit model of the other side. Interestingly, the authors find that the depA* models perform better than the indA* models — showing that training on predicted answers is a stronger signal for building an accurate mental model than just sharing training data. — Overall, the paper is clearly written and thorough, the proposed formulation is novel and makes sense, and results outperform prior work. I’m happy to recommend this for publication. — Minor comment: RL and AQM objectives are indeed closely related, just that model-free RL (or how the game is set up in prior work) models the other side implicitly in its policy / value function. The two approaches seem complementary though — in that RL training can be augmented with these dialog agents having a model of the other side.

Reviewer 2



[Update After Rebuttal] The rebuttal and other reviews address many of the issues I raised. I hope authors improve the readability of the paper for camera ready. I increase my score to 6. [END of Update] This paper proposes a method to ask questions based on the information gain for the task of goal oriented dialog. The paper is terribly hard to follow hindering the communicating the technical contributions of the paper. I do not think that NIPS community would benefit from this paper in its current form. I listed my questions and issues I found with the narrative or nitpicking details about the writing. I hope it would help authors increase the quality of the paper. Q1 Is there a non-trivial reason why you haven't tried your method on Visual Dialogue (VisDual) benchmark? Q2 Similar to papers in the literature, the terms answer and class or target is confusing in the paper. Do you thing replacing respond <> answer, class <> target or ground-truth or gold-label could be better? Q3 L62 states that AQM is a method to understand existing methods. I do not find evidence in the remaining of the paper. Could you clarify that part? Please cite following papers. I also highly recommend reading their related work sections: "Unified Pragmatic Models for Generating and Following Instructions" Fried et. al. "Reasoning about pragmatics with neural listeners and speakers" Andreas et. al. "A joint speakerlistener-reinforcer model for referring expressions" Yu et. al. "Colors in context: A pragmatic neural model for grounded language understanding" Monroe et. al. L1 The very first sentence is very important. However, this one does not any depth to the narrative. I highly suggest removing it. L4 "adequate" -> correct, right ? L6 "owing to" -> due to L7 the theory of mind L17 "classical" subjective, please remove L17 I am not sure "including" is the right word. Maybe "problem that needs to be addressed for digital personal ..." L21 "though a good solution .." this tone is not appropriate and not common in our community. Please rephrase it. L31 the theory of mind L67 the first sentence is not connected to the rest of the paragraph. Please rephrase. L137 "selects" -> select L142 "it is why" --> "thus, we need to train the AQM's questioner" L202 "indA" "depA" "countQ" all the abbreviations are super cryptical. Please remove all of them and expand table 1 with several columns where you indicate different features and learning regimes L204-207 First start motivating the experiments. Why are you doing the experiment? What are the implications of possible outcomes. Then explain the experimental method. L211-213 Please motivate first why you need to introduce counting method. L226 "comparative" --> "competing" "compared" L236 how do you measure interpretability? You did not mention interpretability till now. It is very abrupt. L260-266 with all abbreviations it is impossible to read this part. Please consider rephrasing.

Reviewer 3



Summary: The current work proposes a new framework, Answerer in Questioner’s mind (AQM), for goal-driven visual dialog. Here, questioner has a model of the answer, which it uses to pick a question from a pool of candidate questions, based on information theoretical heuristics. Experiments are shown on (a) MNIST Counting (toy dataset), and (b) GuessWhat?!, showing improvements over competing baselines. Strengths: (S1) The approach is well motivated by the obverter technique, based on the theory of mind. The paper does a good job of putting forth these arguments. Weaknesses: (W1) My primary concern is regarding the limited natural language complexity of the Answerer the Questioner is trying to model. From the datasets used and examples shown, Answerer either utters single word answers (classification in case of MNIST Counting) or binary answers (GuessWhat?!). In my opinion, the proposed approach benefits the most when there is richness in language used by Answerer, to which Questioner adopts itself via this internal modeling. (C1) explores another dimension to this. (W2) The manuscript has a lot of writing errors/typos (few listed at the end). Even though it does not hinder the understanding, it does break the flow of the reader. I urge the authors to carefully proofread their manuscript. Comment: (C1) Modeling Answerer seems more impactful if Answerer has quirks. For instance, in any application, different human users have different preferences, personas, language styles, etc. Such experiments are missing, which would potentially make this paper stronger. As an example, a color blind Answerer (in case of MNIST counting) should drive the Questioner away from asking color based questions, and so on. (C2) From my understanding, the model seems to benefit from both YOLO object candidates and the selection of question using proposed approach. On the other side, competing methods seem to be doing this visual inference themselves. Perhaps, include a baseline that also conditions on these proposals. (C3) Less of a comment and more of a curious question. L141-155 mentions that this approach needs training for a given Answerer. Any thoughts on online training or a low-shot adaptation for a novel answerer? (C4) L117: Functionality of Q-Sampler has not been introduced so far. (C5) L122, Eq2: Typo in the equation. Are the bounds \Pi_{j=1}^{t} for the product? (C6) L169-170: How is 85% -> 46.6%? These two sentences are unclear. (C7) L212: What does consecutive questions mean here? Aren’t these just pool of questions that make up Q-Sampler, not in any particular sequence? (C8) L191: Is the class label from YOLO also used? Typos: L14: Swap closing mark and full stop L44: split -> splits L67: Not sure what the sentence is trying to convey, perhaps a typo somewhere? L102: Typo in the sentence. L115: substituted to -> substituted with / by L115: the equation -> an equation L120: to -> as L137: selects -> select L235: to -> with L235: task training -> task of training